# Molecular Characterization of Diverse Wheat Genetic Resources for Resistance to Yellow Rust Pathogen (*Puccinia striiformis*)

Muhammad Saeed [1,2], Muhammad Ibrahim [3], Waqas Ahmad [3], Muhammad Tayyab [3], Safira Attacha [3], Mudassar Nawaz Khan [4], Sultan Akbar Jadoon [5], Syed Jehangir Shah [3], Shaista Zeb [3], Liaqat Shah [6], Fazal Munsif [7], Ahmad Zubair [8], Jie Lu [1], Hongqi Si [1,9,10,11,*] and Chuanxi Ma [1,9,10,11]

1 School of Agronomy, Anhui Agricultural University, Hefei 230036, China
2 Cereal Crop Research Institute, Pirsabak, Nowshera 24100, Pakistan
3 Institute of Biotechnology and Genetic Engineering, The University of Agriculture, Peshawar 25130, Pakistan
4 Department of Biotechnology and Genetic Engineering, Hazara University, Mansehra 21120, Pakistan
5 Department of Plant Breeding and Genetics, The University of Agriculture, Peshawar 25130, Pakistan
6 Department of Agriculture, Mir Chakar Khan Rind University, Sibi 82000, Pakistan
7 Department of Agronomy, The University of Agriculture Peshawar, Amir Muhammad Khan Campus, Mardan 23200, Pakistan
8 Agricultural Research Institute, Tarnab, Peshawar 24330, Pakistan
9 Key Laboratory of Wheat Biology and Genetic Improvement on South Yellow & Huai River Valley, Ministry of Agriculture, Hefei 230036, China
10 National United Engineering Laboratory for Crop Stress Resistance Breeding, Hefei 230036, China
11 Anhui Key Laboratory of Crop Biology, Hefei 230036, China
* Correspondence: sihq2002@163.com

**Abstract:** Yellow rust (YR) epidemics have affected wheat productivity worldwide. YR resistance (Yr) is eminent in wheat; however, it is continuously invaded by evolving YR pathogen *Puccinia striiformis* (*Pst*.). Understanding the Yr genes' diversity among the available germplasm is paramount to developing YR-resistant cultivars. In this study, 14 wheat genotypes were screened for their relative resistance index (RRI) and Yr genes/QTL via linked microsatellite markers. RRI screening categorized the studied genotypes into susceptible ($<5$; $4.44 \pm 0.75$), moderate ($5$–$7$; $6.11 \pm 0.64$), and resistant ($>7$; $8.45 \pm 0.25$) bulks ($p < 0.001$). Genetic analysis using 19 polymorphic microsatellite markers revealed 256 alleles, which were divergent among the three resistance bulks. Markers Xbarc7 and Xgwm429 showed the highest allelic diversity in comparison to Xbarc181, Xwmc419, SCAR1400, and Xgwm130. Resistant bulk showed associated alleles at Yr18 gene-linked markers Xgwm295, cssfr6, and csLV34. Other RRI-associated alleles at markers Xbarc7 and Xbarc101 showed weak and moderate linkages, respectively, with the Yr5 gene; whereas, a moderate association was noted for the Yr15 gene-linked marker Xgwm11. Marker Xwe173 linked with the Yr26 gene showed associated alleles among the susceptible bulk. Cross combinations of the parental lines forming recombinant inbred lines (RILs) demonstrated net higher RRI implying favorable allelic recombination. These results support reports and field observations on novel *Pst.* races that triggered Yr26, Yr5, and Yr15 busts in recent past. This study further implies that pyramiding all stage resistance genes (Yr5, Yr10, Yr15, and Yr26) with adult plant resistance genes (Yr18 and Yr62) should provide sustained YR resistance. The associated alleles at Yr genes-linked markers provide a basis for marker-assisted YR resistance breeding in wheat.

**Keywords:** yellow rust; relative resistance index; Yr genes; marker assisted breeding; Yr linkage; all stage resistance; adult plant resistance

## 1. Introduction

Production of wheat (*Triticum aestivum* L.), a major staple crop, is endangered by the increasing severity and incidence of yellow rust (YR) disease. YR is an epidemic disease caused by the fungus *Puccinia striiformis* (*Pst.*), infecting both spring as well as winter wheat [1]. YR is one of the most diverse, destructive, and widespread diseases decimating wheat productivity [2]. Around 88% of wheat productions are vulnerable to YR [3]. According to an estimate, every year there is a loss of 5–6 million tonnes of wheat due to YR, at an estimated cost of USD 979 million [3]. Grain-filling is directly impacted by YR, especially if the disease occurs after anthesis. This is because the affected photosynthetic cells are less effective in intercepting light and using radiation, which lowers yields [4]. Efficient dispersal, i.e., the ability of *Pst.* urediospores to travel long distances, even across continents, has resulted in its global spread [5].

Severe yellow rust outbreaks have been reported, ranging in intensity from two percent to complete crop loss [6]. The latest entry to the list of countries affected by the YR epidemic is Zimbabwe [7]. However, the epidemic occurs more frequently (2 or 3 years of every 5 years) in Central and South Asia, China, West Africa, Australasia, the UK, the USA, and South America [8]. Besides the higher dispersability of the YR pathogen, another reason for its widespread presence is the high mutation rate of *Pst.* resulting in its higher adaptability as explained by its complicated life cycle [9]. The sexual stage in the pathogen life cycle hosted by *Berberis vulgaris* plays a key role in generating novel genetic combinations, which results in potential virulent *Pst.* isolates that may lead to epidemics and swift alterations in wheat resistance [10].

With five distinct sporulation stages, *Pst.* has a highly complicated life cycle. The pathogen lifecycle is split into asexual stages, utilizing wheat (the primary host), and sexual cycle utilizing *B. vulgaris* (alternate host; Figure 1) [11]. However, alternate host *Berberis* spp. colonized by *Pst.* are seldom seen in the wild [12,13]. Most of the wheat-growing regions around the world lack the opportunity for *Pst.* to undergo complete genetic recombination (due to the absence of an alternate host) with the exception of the Himalayan region, i.e., China and Pakistan [14–17]. The seasons of the development of *B. vulgaris* and wheat growing in the region are also simultaneous [13].

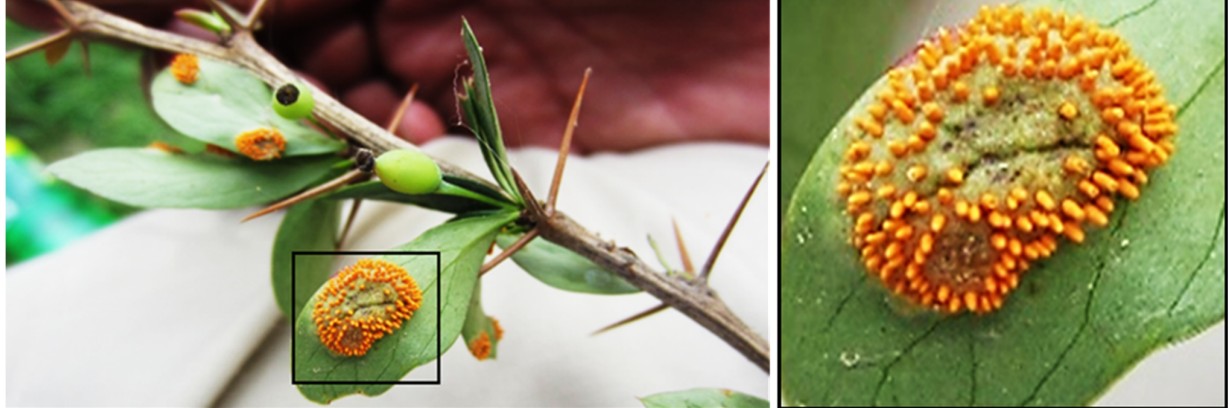

**Figure 1.** Alternate host (*Berberis vulgaris*) of yellow rust pathogen. *B. vulgaris* was found in shuttle wheat breeding station at Kaghan valley of Pakistan (the Himalayan region). Pathogen develops and sexually reproduces in orange-colored aecia, the aeciospores are then spread by wind. (Photo by, Saeed M.).

In recent years, *Pst.* populations have acclimated to hot climates, grown more aggressive, and have become genetically diverse. Control of the *Pst.* and protection against YR primarily involves the use of fungicides and endogenous plant resistance. Fungicides can be used to reduce stripe rust, but their expense and potential environmental impacts raise serious questions [18]. Growing resistant cultivars effectively and sustainably should

control YR [19,20]. Utilization of YR-resistant genotypes to lessen YR damage is the most cost-effective and environmentally friendly strategy because it protects grain production and lowers the demand for fungicides [21]. Widespread interest in the YR resistance discovery has led to the identification of a large number of YR resistance (Yr) and susceptibility genes [22].

Sustainable protection against YR through the employment of resistant wheat cultivars has been complicated by the swift adaptability of the *Pst.* [23,24]. However, using resistance genes individually to prevent yellow rust is no longer effective due to the appearance of *Pst.* pathotypes adapted to a wide range of host resistance (Yr) genes. Genetic resistance conferred by a single gene may be prone to break down as a result of the pathogen's virulence evolution [25]. Since the 1950s, there have been five boom and bust cycles, where otherwise resistant wheat cultivars were devastated by newly adapted *Pst.* Races [26]. Breeding methods are now focusing on genomic and molecular approaches for developing durable YR-resistant wheat cultivars using multigenic approaches [27].

Breeding programs use two types of stripe rust resistance genes, i.e., all-stage resistance (ASR) and adult plant resistance (APR) genes depending on the resistance conferred at different growth stages [28]. The majority of the yellow rust ASR genes discovered are limited by specificity to particular races resistance genes [29]. Race specificity makes germplasm carrying the ASR genes vulnerable and this kind of resistance may be broken down within a short span of time [30]. Therefore, growing cultivars expressing race-specific ASR genes over large geographical areas exert selection pressure on the *Pst.* resulting in the fast evolution of new virulent races [31]. The ASR genes mostly encode for proteins containing nucleotide-binding sites with leucine-rich repeat (NBS-LRR). The NBS-LRR allows specific identification of the avirulence (Avr) proteins by respective sets of R gene products inside the host cells [32]. Examples of Yr genes carrying NBS-LRR include YrSP, Yr5, and Yr10, etc. [33].

In contrast to the ASR, the APR expresses in late plant growth stages, resulting in slow rusting phenotypes [11]. The APR resistance may also be regulated by temperature, a phenomenon known as high-temperature adult plant (HTAP) resistance. APR is usually partial, and may or may not be sufficient to resist the *Pst.* invasion [20,34]. However, the key positive dimension of the APR is that this resistance type is race non-specific and is considered more durable and effective against many *Pst.* pathotypes. For example, Yr18 has remained durable and effective against YR for the longest duration [35,36]. To boost resistance level and durability, there is a need to combine APR with ASR [37].

Screening for the Yr genes and associated quantitative traits loci (QTL) requires the use of molecular approaches such as genome-wide association studies (GWAS) and more frequently used associated DNA markers. DNA markers among others include microsatellite or simple sequence repeats (SSR), random amplified polymorphic DNA (RAPD), restriction fragments length polymorphism (RFLP), and single nucleotide polymorphism (SNPs), etc. [38,39]. DNA markers are needed to construct genetic maps, which may be utilized to find the location of Yr genes or Yr QTL [40]. By means of marker-assisted selection (MAS), these Yr QTL may be introduced into the cultivated wheat genotypes to develop resistant varieties [41]. Furthermore, markers that are frequently associated with important genes may serve as a starting point for map-based transgenic germplasm development [42].

Allelic variations at different microsatellite markers were used to screen genetic differences among susceptible, moderately resistant, and resistant wheat genotypes as indicated by their low, moderate, and high relative resistance index (RRI), respectively. We used different analytical tools to identify novel microsatellite alleles associated with YR resistance. The associated microsatellite markers (of the total 19 markers) were further mined for their proximity with the known Yr genes/QTLs (Yr18, Yr26, Yr5, Ye15, Yr62) in the GrainGenes database [43]. The recombinant inbred lines (RILs) developed using the diverse parents were further screened based on their RRI. Data from the study are envisioned to form the basis for sustainable YR-resistant breeding in wheat.

## 2. Materials and Methods

This study was conducted at two locations, i.e., The Center of Excellence, Cereal Crops Research Institute (CCRI), Pirsabak, Nowshera, and shuttle wheat breeding station (SWBS) Kaghan, Khyber Pakhtunkhwa Pakistan during the years 2018–2019, 2019–2020, and 2020–2021. CCRI is situated 288 m (945 ft) AMSL at the left bank of Kabul River, 3 km east of Nowshera, at 32° N latitude and 74° E longitude. Nowshera has a warm to hot, semi-arid, and sub-tropical climate with a mean annual rainfall of about 364 mm. It is a hot spot for yellow rust development due to its conducive environment. At 34°50′ N and 73°31′ E, Kaghan is a mountain valley with an elevation of 650 m (2134 ft) above sea level. Due to its favorable summer climate for wheat, this shuttle breeding site was built to promote wheat reproduction. With an average humidity of 59%, summertime temperatures typically range from 20 °C to 26 °C.

### 2.1. Germplasm Source, Matting Design and Sowing Layout

A diverse panel of wheat genotypes including fourteen parents (9 female lines and 5 male testers) and 45 recombinant inbred lines (RILs), resulted from lines × testers crossings, were evaluated under YR-stress conditions. Parental germplasm was procured from three wheat research groups, i.e., (i) two advance lines from State Key Laboratory, department of crop breeding and genetics, Anhui Agricultural University Hefei, China, (ii) five lines from International Maize and Wheat Improvement Center (CIMMYT) Mexico, and (iii) seven from Wheat Breeding Section of Cereal Crops Research Institute Pirsabak Pakistan [44].

Hybridization (cross combinations of lines × testers) of breeding material (Figure 2) was carried out at the wheat breeding section, CCRI Pirsabak, during the winter cropping season 2018–2019. Filial generation advancement was done with two generations per year at SWBS Kaghan (June to October), and CCRI Pirsabak (November to May) from 2019 to 2021. The initial selection of the RILs and the subsequent generation advancement were based on screening for YR-resistant phenotypes. The RRI data were ultimately collected on the F6 generation of the RILs. The selected 45 F6 RILs (selected resistant plants from the F5 population) along the respective parents (9 lines and 5 testers) were sown at CCRI Pirsabak in the 1st week of November 2021 using randomized complete block design, with three replications. Each plot had 4 rows of 5 m length with 0.25 m row-to-row distance and 0.6 m plot-to-plot distance. The experimental plots were fenced by a highly susceptible wheat genotype (Morocco) to encourage YR disease development. Standard agronomic practices were carried out as per the requirement of the wheat crop. The fertilizer was applied in split doses (half at sowing and the remaining half at the early boot stage).

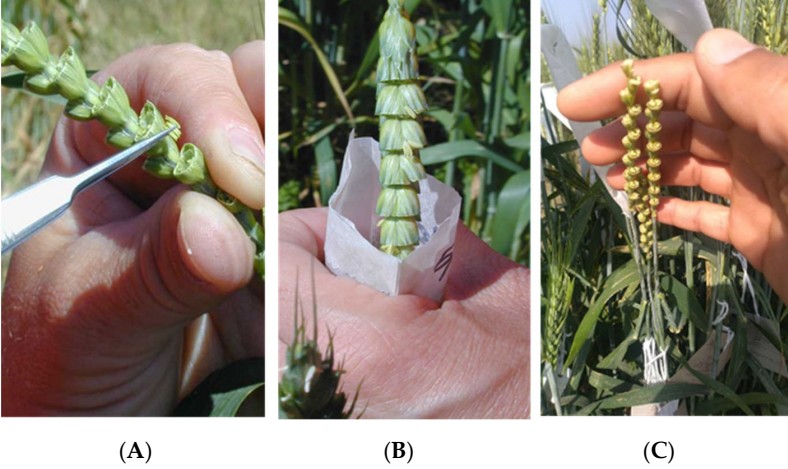

(**A**)　　　　　　　　(**B**)　　　　　　　　(**C**)

**Figure 2.** Hybridization of diverse wheat lines to development of recombinant inbred lines. The picture shows emasculation (**A**), crossing (**B**), and successful and healthy seed setting (**C**) of hybrid seeds. (Photo by, Saeed M.).

### 2.2. Inoculation of Breeding Germplasm

Experimental material was inoculated at the vegetative growth (booting) stage with a fresh inoculum of the dominant Pakistani pathotype (*Pst.* 574232) [27]. The YR-inoculum was obtained from the Crop Diseases Research Institute, National Agricultural Research Centre (NARC), Islamabad. The uredospore's suspension was prepared using 0.1 g of YR inoculum in 1 L of distilled water and added a few drops of Tween-20 for efficient germination of spores and fungal hyphae penetration. At sunset, each experimental plot was inoculated uniformly with the suspension by using a turbo-air sprayer.

### 2.3. YR-Disease Severity and Host Response

YR disease reaction and severity in the host plant were determined based on the percent flag leaf area affected. Peterson et al.'s [45] method based on the visual determination of the proportion (1 to 100%) of leaf area infected by yellow rust was utilized to assess disease severity using a modified Cobb scale. The response of the host was detected by its reaction against the disease. The data on host responses and disease value were recorded and presented as, "I" denotes fully immune type by having no yellow rust infection, "R" denotes resistant by having observable with no YR pustules and necrosis on leaf, "MR" denotes moderately resistant by showing small and tiny pustules with little necrosis on leaf, "MS" denotes moderately susceptible by having an intermediate level of pustules and zero necrosis but having observable chlorosis on leaf, and "S" denotes fully susceptible with fresh and bulky pustules with necrosis and chlorotic area on leaf surface.

### 2.4. Disease Assessment and Host Phenotyping

Yellow rust response of the tested wheat genotypes was characterized using two epidemiological parameters: average coefficient of infection (ACI) and RRI. The highest ACI of the tested candidate lines was set at 100 and all other lines were accustomed accordingly to calculate the country average relative percentage attack (CARPA). Using a 0 to 9 scale previously designated as resistance index (RI) was re-designated as relative resistance index (RRI). From CARPA the value of RRI was calculated on the 0 to 9 scale, where 0 denotes the most susceptible and 9 highly resistant [46]. The RRI was calculated as "RRI = $(100 - \text{CARPA})100 \times 9$". The susceptible (RRI < 5), moderate (RRI = 5–7), and resistant (RRI > 7) categorization was performed using the protocol of Aslam [47].

### 2.5. YR-Coefficient of Infection

Yellow rust coefficient of infection (CI) was noted by multiplying prevailing disease severity with host hypersensitive response, using the protocol of Pathan and Park [48]. CI was calculated by multiplying the disease infection severity value by its host reaction as follows.

$$\text{Coefficient of infection} = \text{Disease severity} \times \text{Infection type}.$$

The disease severity is the visual observation of the percent leaf area affected by YR as explained above. Infection type was 1.0 for susceptible (S) host response, 0.8 for moderately susceptible (MS), 0.6 for moderately resistant (MR), and 0.2 for resistant (R). The ACI was the mean CI of replicates for each experimental plot.

### 2.6. DNA Extraction

The parental genotypes were sampled for DNA and subjected to microsatellite analysis. Fresh young leaf samples were collected from the field in pre-labeled plastic bags and immediately placed on dry ice in a Styrofoam box. A total of 42 plants were sampled for leaves (3 replicates for each of the 14 parental genotypes). DNA was extracted from fresh leaf samples in the laboratory at the Institute of Biotechnology and Genetic Engineering, The University of Agriculture, Peshawar using the CTAB method [49].

### 2.7. Polymerase Chain Reaction and Genotyping

Microsatellite markers were amplified in a thermal cycler (T100; Bio-Rad, Hercules, CA, USA). Each reaction was 25 μL with 12.5 μL of 2X DreamTaq Green PCR master mix (K1081, Thermo fisher scientific), 8.5 μL double distilled $H_2O$, 1 μL each of forward and reverse primers (Table 1). A touchdown PCR method was developed to increase PCR specificity. The reaction was run with an initial denaturation at 95 °C for 5 min, followed by 10 repeat cycles (denature at 94 °C for 45 s, primer annealing at 64 °C for 45 s (with a gradual decrement of 1 °C at every cycle), and extension at 72 °C for 45 s. This was followed by another repeat of 25 cycles with a thermal profile of denaturation at 94 °C for 45 s, annealing at 54 °C for 45 s, and extension at 72 °C for 45 s; followed by one last extension cycle at 72 °C for 10 min. In control reactions, sterile distilled water was replaced with template DNA to check for contamination.

**Table 1.** Microsatellite primers used in the current study, their sequences, and annealing temperatures.

| Primer Name | Primer Sequences | | Annealing Temp °C |
| --- | --- | --- | --- |
| | **Forward** | **Reverse** | |
| Xgwm11 | GGATAGTCAGACAATTCTTGTG | GTGAATTGTGTCTTGTATGCTTCC | 52 |
| Xgwm295 | GTGAAGCAGACCCACAACAC | GACGGCTGCGACGTAGAG | 54 |
| Xgwm140 | ATGGAGATATTTGGCCTACAAC | CTTGACTTCAAGGCGTGACA | 53 |
| S23M41275 | TCAACGGAACCTCCAATTTC | AGGTAGGTGTTCCAGCTTGC | 55 |
| csLV34 | GTTGGTTAAGACTGGTGATGG | TGCTTGCTATTGCTGAATAGT | 54 |
| csLVMS1 | CTCCCTCCCGTGAGTATATTC | ATCAAAATCCCATTGCCTGAC | 54 |
| Xbarc151 | TGAGGAAAATGTCTCTATAGCATCC | CGCATAAACACCTTCGCTCTTCCACTC | 53 |
| Xbarc181 | CGCTGGAGGGGGTAAGTCATCAC | CGCAAATCAAGAACACGGGAGAAAGAA | 52 |
| Xwe173 | F;GGGACAAGGGGAGTTGAAGC | GAGAGTTCCAAGCAGAACAC | 56 |
| Xwmc419 | GTTTCGGATAAAACCGGAGTGC | ACTACTTGTGGGTTATCACCAGCC | 57 |
| Xgwm191 | F;AGACTGTTGTTTGCGGGC | TAGCACGACAGTTGTATGCATG | 55 |
| Xgwm501 | GGCTATCTCTGGCGCTAAAA | TCCACAAACAAGTAGCGCC | 54 |
| Xbarc7 | CGCCATCTTACCCTATTTGATAACTA | TTGTACATTAAGTTCCCATTA | 55 |
| Xgwm429 | TTGTACATTAAGTTCCCATTA | TTTAAGGACCTACATGACAC | 57 |
| cssfr6 | CTGAGGCACTCTTTCCTGTACAAAG | GCATTCAATGAGCAATGGTTATC | 55 |
| Xbarc101 | GCTCCTCTCACGATCACGCAAAG | GCGAGTCGATCACACTATGAGCCAATG | 61 |
| SCAR1400 | CACTCTCCTCAAACCTTGCAAG | CACTCTCCTCCACTAACAGAGG | 56 |
| Xgwm 130 | AGCTCTGCTTCACGAGGAAG | CTCCTCTTTATATCGCGTCCC | 53 |
| Xgwm 251 | CAACTGGTTGCTACACAAGCA | GGGATGTCTGTTCCATCTTAG | 59 |

The PCR products were resolved on 2% agarose gel along with a 1 kb DNA ladder. The gels were manually scored for the presence and absence (0, 1) of alleles of different sizes for each microsatellite locus. The data were set in a matrix of excel sheets for further analysis.

### 2.8. Statistical Analysis

The ACI data were used in computer packages MS Excel and Statistix 10.0 for analysis of variance among genotypes. The overall comparison of individual means was calculated by least significant differences (LSD) at a 0.05% probability level. The data were then subjected to the calculation of RRI from the ACI values as explained above. Regression among RRI and ACI was computed by plotting the two variables on a scatter plot using MS Excel.

Means and standard errors for RRI among different parents and RILs, groups of parents (susceptible, moderate, and resistant), and among alleles of significantly associated genotypes were calculated using SPSS software version 23.0. One-way ANOVA along with post hoc Tukey's test was performed for pair-wise comparisons.

The allelic data of microsatellite markers were analyzed using POPGENE and STRUC-TURE software for genetic diversity parameters. Allelic polymorphism and allele frequencies at the microsatellite markers were calculated for susceptible, moderate, and resistant

parental groups using POPGENE software. The total number of alleles shared and unique alleles were calculated for each parental group. The microsatellite data were further evaluated for ancestral admixture among the three parental groups based on correlated allele frequencies using STRUCTURE software v2.3.4.

Association between microsatellite allelic polymorphism and RRI was performed using principal component analysis (PCA) in SPSS software. The analysis was based on correlations among RRI and different alleles at each microsatellite marker. Varimax rotation with a convergence of 25 iterations was used for constructing the PCA plot.

## 3. Results

### 3.1. Variable Response of Genotypes against Yellow Rust

Analysis of variance showed that the tested genotypes (including parents and RILs) were significantly different ($p < 0.01$) for yellow rust ACI (Table 2). Among parents, lines demonstrated significant variation ($p < 0.05$), while the testers exhibited highly significant variability ($p < 0.01$) for YR ACI. The interaction between lines and testers was non-significant. Parents vs. RILs contrast, which is the comparison of the overall mean of parents and RILs, was highly significant ($p < 0.01$) for YR ACI.

**Table 2.** Analysis of variance for yellow rust average coefficient of infection (ACI) in parental lines and RILs evaluated at CCRI Pirsabak, Nowshera.

| SOV | DF | YR- Average Coefficient of Infection | $f$ Value |
| --- | --- | --- | --- |
| Replications | 2 | 29.9 | 1.1[ns] |
| Genotypes | 58 | 359.7 | 13.8 ** |
| Parents | 13 | 453.2 | 17.4 ** |
| Lines (L) | 8 | 212.8 | 8.2 * |
| Tester (T) | 4 | 1817.8 | 69.6 ** |
| L × T | 32 | 174.3 | 6.7 [ns] |
| RILs | 44 | 330.7 | 12.7 ** |
| Parents vs. RILs | 1 | 420.6 | 16.1 ** |
| Error | 116 | 26.1 | |
| CV(%) | – | 14.2 | |

** Significant at 1% probability level; * significant at 5% probability level; [ns] non-significant.

### 3.2. Parental Line Response against Yellow Rust

Disease reaction on wheat genotypes was evaluated from ACI and subsequently the plants' response was detected from the RRI values. Based on the disease status the genotypes were grouped into three resistance classes, i.e., susceptible (S), moderate (M), and resistant (R) (Table 3). Of the total parents, three lines (PR125, PR127, and PR128) and two testers (WD17 and KS17) were marked resistant. A wide range of disease responses was observed among the diverse germplasm ranging from 5R to 40S. ACI of parental genotypes ranged from 2.3 to 35.2%. The resistant genotypes showed a low ACI (2.7–5.2). ACI 11–25 was observed for moderately resistant genotypes, whereas susceptible genotypes with low levels of resistance showed an ACI of 26–30.

Similarly, the RRI of parent genotypes ranged from 4.1 to 8.7. Significant pair-wise differences were observed among the three resistance classes. Maximum RRI values (8.2–8.7) were depicted by the resistant parental genotypes, which were significantly higher than the genotypes in the susceptible and moderate bulks. The minimum RRI values (4.0–4.5) were noted in lines AN179 and PR130 and tester PK15 grouped in susceptible bulk.

**Table 3.** Response of wheat parents to yellow rust infections.

| Parents | Disease Status | Disease Response * | ACI | CARPA | RRI |
|---|---|---|---|---|---|
| PR123 | M | 30MS | 21.3 | 33.7 | 6.0 ± 0.16 [b] |
| PR125 | R | 10MR | 3.7 | 5.8 | 8.3 ± 0.13 [c] |
| PR126 | M | 40M | 22.0 | 34.8 | 5.6 ± 0.40 [ab] |
| PR127 | R | 10MR | 4.7 | 7.4 | 8.2 ± 0.06 [c] |
| PR128 | R | 5R | 2.7 | 4.2 | 8.5 ± 0.03 [c] |
| PR129 | S | 30MSS | 24.1 | 38.0 | 4.8 ± 0.36 [ab] |
| PR130 | S | 30MSS | 28.2 | 44.5 | 4.3 ± 0.43 [a] |
| AN179 | S | 30S | 34.3 | 54.2 | 4.0 ± 0.70 [a] |
| AN837 | M | 30M | 18.3 | 29.0 | 6.4 ± 0.26 [b] |
| PS13 | M | 35MS | 24.3 | 38.4 | 5.5 ± 0.06 [ab] |
| PS15 | M | 30M | 15.3 | 24.2 | 6.9 ± 0.20 [bc] |
| PK15 | S | 40S | 35.2 | 55.6 | 4.5 ± 0.13 [ab] |
| KS17 | R | 5R | 2.3 | 3.7 | 8.7 ± 0.03 [c] |
| WD17 | R | 10R | 5.2 | 8.2 | 8.5 ± 0.26 [c] |

Key: ACI = average coefficient of index, RRI = relative resistance index, CARPA = country average relative percent attack, S = susceptible, M = moderate, R = resistant. Superscripts (a, b, c) on the RRI values show pair-wise significant differences among genotypes ($p < 0.01$). * = disease response scale based on Akhtar et al., 2002 [46].

### 3.3. Regression between Resistance Indices

Regression analysis revealed a strong relationship between RRI and ACI among tested genotypes (Figure 3). A linear negative correlation between RRI and ACI was apparent with a good fit ($R^2 = 0.99$), implicating that wheat parents were influenced due to the high infection rate in plants by indicating lower RRI. The regression equation $y = -0.1414x + 9.061$ showed that with a 14% increase in ACI, the RRI of the wheat genotypes decreased by 9%.

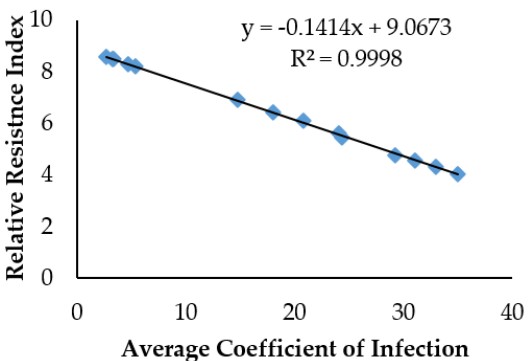

**Figure 3.** Regression analysis between relative resistance index (RRI) and average coefficient of infection (ACI) for yellow rust in parental wheat.

### 3.4. Parental Genotype Bulks Based on RRI

Based on RRI differences, parental genotypes were divided into three bulks denoted as susceptible (RRI = 4.0–5.6), moderate (RRI = 5.5–6.8), and resistant (RRI = 8.3–8.7) (Figure 4). The susceptible bulk included four genotypes (PR129, PR130, AN179, and PK15), five genotypes (PR123, PR126, AN837, PS13, and PS15) were grouped in moderate bulk, and five genotypes (PR125, PR127, PR128, KS17, and WD17) having high RRI made the resistant bulk. The mean RRI was 4.44 ± 0.75 for susceptible bulk, 6.11 ± 0.64 for moderate bulk, and 8.45 ± 0.25 for resistant bulk. The RRI of each group was significantly different from the others ($p < 0.001$).

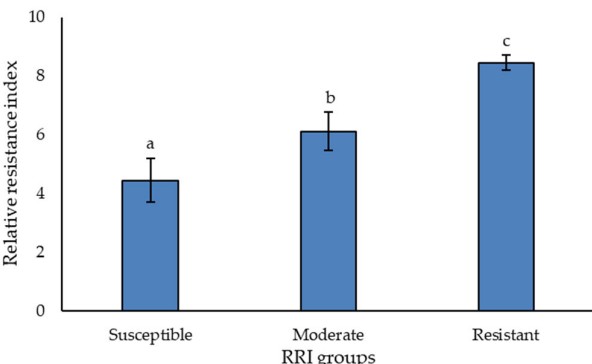

**Figure 4.** Grouping of parental genotypes into susceptible, moderate, and resistant bulks based on RRI. Error bars indicate standard deviation. Letters (a, b, and c) shows differences among groups ($p < 0.05$).

*3.5. Microsatellite Polymiorphism between Resistance Bulks*

Microsatellite analysis revealed differences between the three RRI-based bulks of parental genotypes. A total of 256 alleles were observed at 19 microsatellite markers across the 14 parental genotypes. Polymorphism in the number of alleles at different microsatellite markers was observed among susceptible, moderate, and resistant bulks (Table 4). Minor differences in alleles per locus were observed among moderate and resistant bulks compared to susceptible bulk. Higher differences in the number of alleles among the three resistance groups were observed at markers Xbarc7 and Xgwm429. Whereas, markers Xbarc181, Xwmc419, SCAR1400, and Xgwm130 showed a similar number of alleles in each bulk. Representative gel images of four highly polymorphic markers having an association with RRI are shown in Figure 5.

**Table 4.** Allelic polymorphism among YR susceptible, moderate, and resistant bulks.

| Marker | Total Alleles | | | | Shared Alleles | | | | Unique Alleles | | |
|---|---|---|---|---|---|---|---|---|---|---|---|
| | T | S | M | R | SM | SR | MR | SMR | S | M | R |
| Xgwm11 | 12 | 6 | 9 | 10 | 4 | 6 | 7 | 4 | | 2 | 1 |
| Xgwm295 | 15 | 9 | 10 | 11 | 6 | 6 | 7 | 4 | 1 | 1 | 2 |
| Xgwm140 | 16 | 13 | 11 | 11 | 9 | 10 | 9 | 9 | 3 | 2 | 1 |
| S23M4127 | 17 | 10 | 12 | 13 | 8 | 7 | 9 | 7 | 1 | 2 | 3 |
| csLV34 | 19 | 10 | 11 | 15 | 6 | 8 | 8 | 5 | 1 | 2 | 4 |
| csLVMS1 | 16 | 13 | 10 | 10 | 9 | 7 | 7 | 6 | 3 | 1 | 2 |
| Xbarc151 | 12 | 10 | 10 | 11 | 8 | 9 | 9 | 7 | | | |
| Xbarc181 | 8 | 8 | 6 | 5 | 6 | 5 | 5 | 5 | 2 | | |
| Xwe173 | 18 | 10 | 14 | 15 | 7 | 7 | 13 | 6 | 2 | | 1 |
| Xwmc419 | 7 | 6 | 6 | 6 | 6 | 5 | 5 | 5 | | | 1 |
| Xgwm191 | 17 | 12 | 14 | 14 | 10 | 11 | 11 | 9 | | 2 | 1 |
| Xgwm501 | 10 | 7 | 8 | 8 | 5 | 5 | 6 | 3 | | | |
| Xbarc7 | 22 | 14 | 19 | 12 | 11 | 7 | 11 | 6 | 2 | 3 | |
| Xgwm429 | 18 | 15 | 10 | 4 | 7 | 4 | 2 | 2 | 6 | 3 | |
| cssfr6 | 14 | 12 | 14 | 14 | 12 | 12 | 14 | 12 | | | |
| Xbarc101 | 9 | 5 | 7 | 7 | 4 | 4 | 6 | 4 | 1 | 1 | 1 |
| SCAR1400 | 5 | 5 | 5 | 5 | 5 | 5 | 5 | 5 | | | |
| Xgwm130 | 6 | 6 | 6 | 6 | 6 | 6 | 6 | 6 | | | |
| Xgwm251 | 15 | 14 | 8 | 13 | 7 | 13 | 6 | 6 | | 1 | |
| Total | 256 | 185 | 190 | 190 | 136 | 137 | 146 | 111 | 22 | 20 | 17 |

Key: YR = yellow rust, T = total number of alleles, S = susceptible, M = moderate, R = resistant. For the number and frequency of alleles, see Table S1.

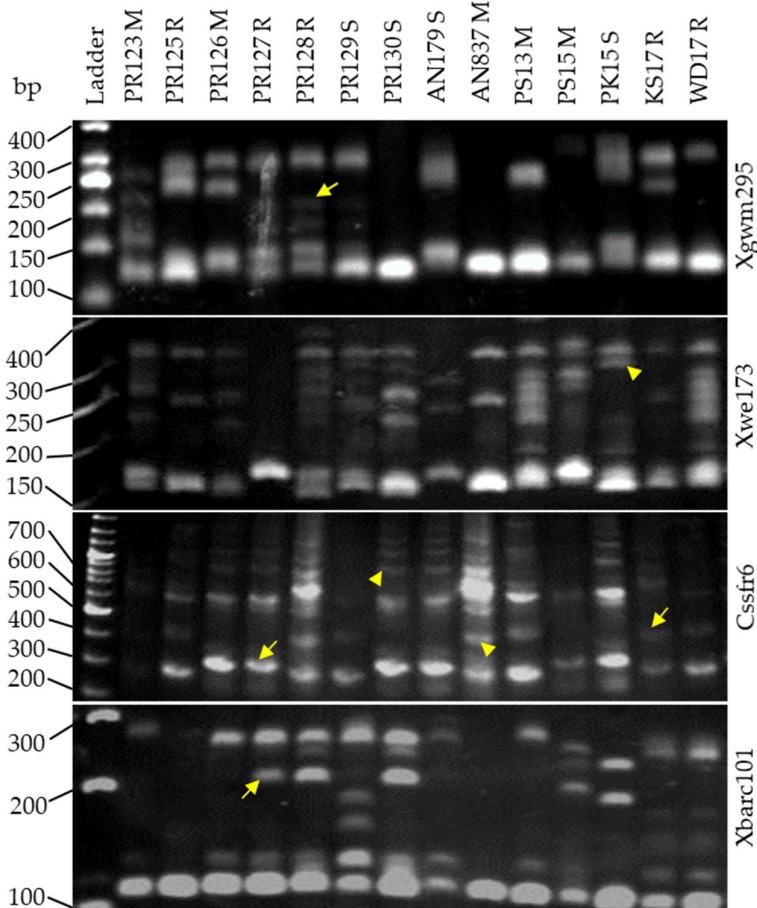

**Figure 5.** DNA microarray obtained using microsatellite markers associated with YR relative resistance index. Variation can be seen among susceptible (S), moderate (M), and resistant (R) genotypes. One kilobase DNA ladder was run as reference. DNA bands indicated by arrow have significant ($p < 0.05$) positive correlation with RRI. DNA bands indicated by arrowhead have significant ($p < 0.05$) negative correlation with RRI.

Susceptible genotypes bulk shared a similar number of alleles (136 and 137, respectively) with moderately resistant and resistant bulk; comparatively, a higher number of alleles (146) were shared among the moderate and resistant bulks. In addition, the unique alleles were more frequent in susceptible bulk, and less in resistant bulk (Table 4). Out of the 19 microsatellite markers, 14 showed at least one unique allele in one of the three resistance bulks. A total number of 59 unique alleles were identified among the three resistance bulks. The highest number of unique alleles (9) was observed at marker Xgwm429, of which six unique alleles were in the susceptible bulk and three were in the moderate bulk. A higher number of unique alleles in the resistant bulk was observed at marker csLV34-F.

### 3.6. Unique Allele Frequencies

The frequencies of unique alleles found at 14 microsatellite markers are presented in Table 5. Of the total 59 unique alleles, ten alleles at seven microsatellite markers showed frequencies higher than 0.2, suggesting a significant association of these alleles with YR resistance. Allele-16 in the susceptible bulk at marker Xwe173 was the most frequent unique allele with a frequency of 0.5. The presence of these alleles at a higher frequency in the susceptible bulk shows a negative association with YR resistance. The remaining nine highly frequent unique alleles were found in moderately resistant and resistant bulks, showing their positive association with YR resistance. The highly associated alleles found at a higher frequency only in the resistant bulk were at markers Xgwm295, Xbarc101, csLV34, and Xgwm11.

**Table 5.** Unique alleles' frequencies among YR susceptible, moderate, and resistant bulks.

| Marker | Allele# | S | M | R | Marker | Allele# | S | M | R |
|---|---|---|---|---|---|---|---|---|---|
| Xgwm191 | 6 | | 0.106 | | Xgwm251 | 10 | | 0.225 | |
| | 9 | | 0.106 | | Xwe173 | 1 | | | 0.106 |
| | 10 | | | 0.106 | | 9 | 0.134 | | |
| Xgwm295 | 6 | 0.134 | | | | 16 | 0.500 | | |
| | 7 | | 0.106 | | Xbarc101 | 4 | | | 0.225 |
| | 11 | | | 0.106 | | 8 | 0.134 | | |
| | 14 | | | 0.225 | | 9 | | 0.106 | |
| Xgwm140 | 4 | | | 0.106 | Xbarc7 | 4 | | 0.225 | |
| | 5 | 0.134 | | | | 11 | | 0.106 | |
| | 6 | 0.134 | | | | 13 | 0.134 | | |
| | 10 | | 0.106 | | | 16 | | 0.225 | |
| | 13 | 0.134 | | | | 21 | 0.134 | | |
| | 14 | | 0.225 | | S23M4127 | 1 | | | 0.106 |
| csLV34 | 2 | | | 0.225 | | 3 | | | 0.106 |
| | 6 | | 0.106 | | | 9 | 0.134 | | |
| | 7 | | | 0.106 | | 12 | | 0.106 | |
| | 10 | | 0.106 | | | 13 | | | 0.106 |
| | 16 | | | 0.106 | | 15 | | 0.106 | |
| | 17 | | | 0.106 | csLVMS1 | 1 | | 0.106 | |
| | 18 | 0.134 | | | | 2 | 0.134 | | |
| Xgwm429 | 1 | | 0.106 | | | 11 | 0.134 | | |
| | 2 | 0.134 | | | | 12 | | | 0.106 |
| | 3 | 0.134 | | | | 13 | 0.134 | | |
| | 4 | 0.134 | | | | 14 | | | 0.106 |
| | 11 | 0.293 | | | Xbarc181 | 3 | 0.134 | | |
| | 12 | | 0.106 | | | 5 | 0.134 | | |
| | 14 | 0.134 | | | Xwmc419 | 5 | | | 0.106 |
| | 16 | | 0.106 | | Xgwm11 | 9 | | 0.106 | |
| | 18 | 0.134 | | | | 11 | | | 0.225 |
| | | | | | | 12 | | 0.225 | |

Key: S = susceptible, M = moderate, R = resistant. the "Allele#" column indicates the alleles' number that was unique in one of the three bulks. The total number of alleles at each marker and their respective sizes are presented in Table S1.

### 3.7. Genetic Diversity Based on Microsatellite Markers

Structure analysis revealed genetic admixture among the three resistance bulks under different numbers of inferred clusters (K) (Figure 6). Origin-based clustering of the microsatellite data was observed at K = 2, separating testers from lines. Genotypes in the susceptible bulk showed an isolated structure under both K = 2 and K = 3. A large proportion of the genotypes (72.2%) in the susceptible bulk showed a high probability in the same inferred cluster. A higher heterogeneity was observed in moderately resistant and resistant bulks showing ancestral admixture in these bulks.

The dendrogram constructed based on the microsatellite polymorphism clearly distinguished testers from lines (Figure 7). This provided enough information about the genetic diversity in the selected gene pool and the basis for further establishing lines × testers for generating RILs. Genetic diversity was also evident among different genotypes of lines and testers. Susceptible line AN179 and resistant line PR127 were the most distantly related to other lines and closely related to each other. This suggests the presence of important

resistance alleles in these two lines, which became masked in the AN179 parent. Later in the RILs, the AN179 cross with the resistant KS17 tester outperformed. The grouping in the dendrogram was based on the entire allelic polymorphism and genetic similarity. No resistance-based grouping was observed, suggesting that only a few alleles are responsible for resistance against YR in the current data set.

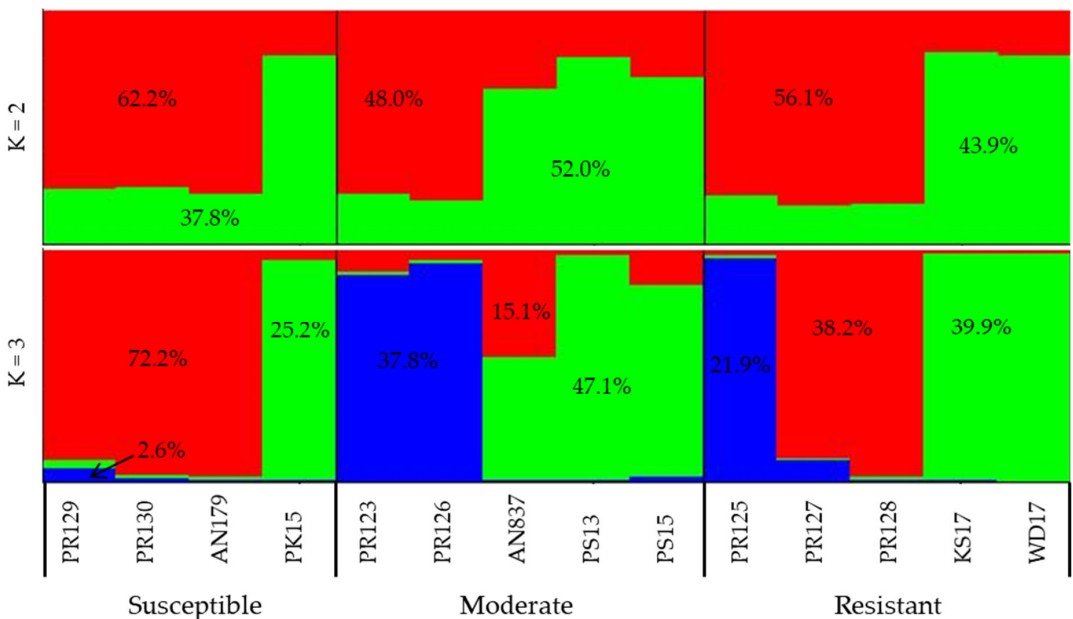

**Figure 6.** Grouping of the three resistance bulks in inferred clusters based on allelic similarities through structure analysis. Different colors represent inferred clusters based on allele frequency differences within the data. The percentage values show amount of the genotypic contribution of the inferred cluster to each resistance bulk.

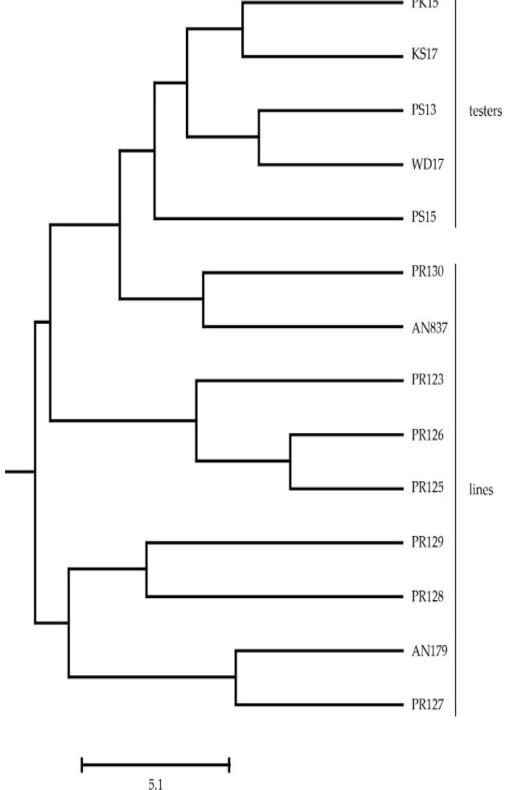

**Figure 7.** Dendrogram showing microsatellite based genetic diversity in the selected wheat gene pool.

### 3.8. Allelic Association with RRI

Principal component analysis grouped the microsatellite alleles as two main clusters (Figure 8A). Of the total 256 microsatellite alleles, 46% alleles were positively correlated with RRI, while the remaining 54% were negatively correlated. Among the positively associated alleles, four microsatellite alleles showed a significantly high association with RRI with a correlation score >0.5. Similarly, the correlation of four alleles was less than −0.5 with RRI showing a significant negative association. Most significant associations with RRI were observed at four different alleles of marker cssfr6, suggesting the role of this marker in conferring YR resistance in wheat. Mean RRI values showed significant differences with the presence and absence of the associated alleles (Figure 8B). For the positively associated alleles, the genotypes in which the allele was present showed significantly high RRI compared to those where the allele was absent. In contrast, for negatively associated alleles, higher RRI was observed in the absence of the allele. It should be noted that allele-4 at marker Xbarc101 (positively associated) was unique in resistant bulk, while allele-16 at marker Xwe173 (negatively associated) was explicitly found in susceptible bulk (Table 5). The remaining highly associated alleles were found in all resistance bulks with varying frequencies (Supplementary Table S1).

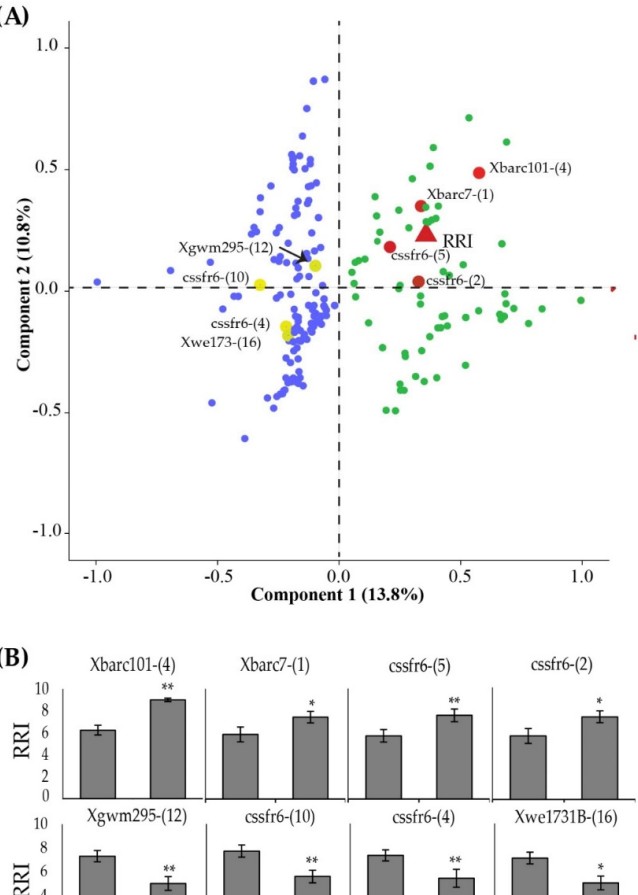

**Figure 8.** Principal component analysis of microsatellite alleles association with RRI. (**A**) Plot showing distribution of microsatellite alleles in relation to RRI. Green cluster shows positively associated alleles; blue cluster shows negatively associated alleles with RRI. Significant positive and negative associated alleles are labeled with marker name and allele numbers (in brackets), and depicted as yellow and red dots, respectively. (**B**) Mean RRI at the identified highly associating alleles. * = $p < 0.1$, ** = $p < 0.05$.

### 3.9. Linkage of Microsatellite Markers with Yr Genes

Based on parental genotypes assessed for YR RRI, and data from 19 microsatellite markers, different markers were found to be closely linked with various APR and ASR Yr genes (Figure 9). The map data were acquired from the GrainGenes database. On chromosome 1B, four microsatellite markers were found to have close proximity with three Yr genes. Marker Xgwm11 was closely linked with Yr15 at an interval of 2.0 cM. Markers Xwe173 and Xbarc181 were on either side flanking the Yr26 genes, while marker Xgwm140 was at a distance of 12.9 cM from the YR29 gene on chromosome 1B. None of the markers used in this study had close proximity to the Yr10 gene. On chromosome 2B, two markers Xgwm501 and Xgwm191 were closely located on either side of the Yr5 gene with intervals of 1.6 and 3.0 cM, respectively. Marker Xgwm251 on chromosome 4B was close to the Yr62 gene. On chromosome 5A, marker Xbarc151 was in close proximity to Yr-associated allele Tb5088. On chromosome 7D, marker cssfr6 shared the same locus with Yr18 at position 65 cM. Two other markers, Xgwm295 and csLVMS1, were also in close proximity to Yr18. Yr33 was distantly located with markers Xgwm130 and csLV34 on chromosome 7D.

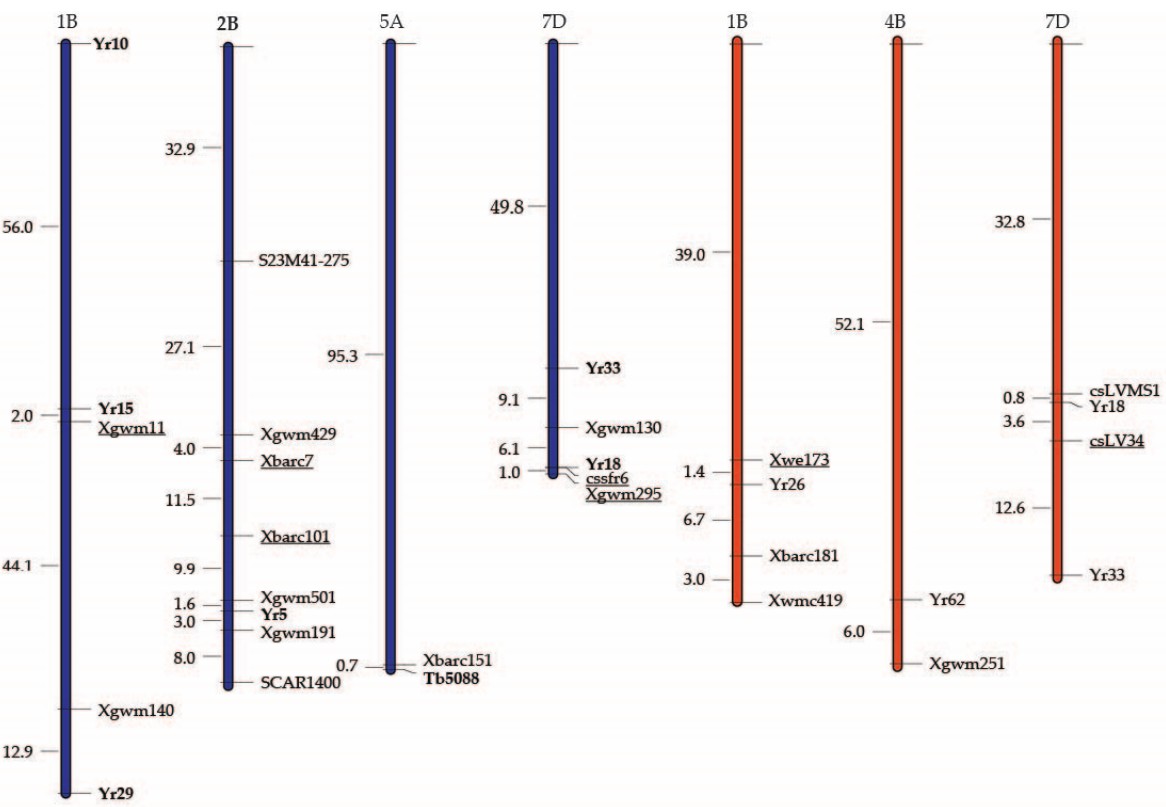

**Figure 9.** Linkage map of microsatellite markers on the respective chromosomes. Locations of Yr genes are depicted in bold. Underlined markers showed association with RRI. The numbers on the left of the chromosomes show marker intervals in centimorgan. Blue color maps indicate markers location based on data from two QTL maps, i.e., wheat consensus 2004 [50], SSR and wheat composite 2004 SSR. The red-colored maps are based on data from QTL map wheat, Yr genes, and QTL 2015 [51].

### 3.10. RILs Response against Yellow Rust

The representative RILs from all the cross combinations of parental lines with testers showed an increase in the frequency of genotypes with higher RRI (Figure 10). Overall, the mean RRI for the parental lines was 6.47 ± 1.73, which was lower than the mean RRI for RILs (7.27 ± 1.65). A higher percentage (28.6%) of parental genotypes showed an RRI value of 6.0, which was replaced by an RRI value of 8.0 for 51% of RILs. The measure of skewness was 0.66 for the parental RRI frequency distribution curve, which increased to

1.66 for RIL. This suggests a shift from normal distribution to a higher proportion of RIL genotypes with high RRI values.

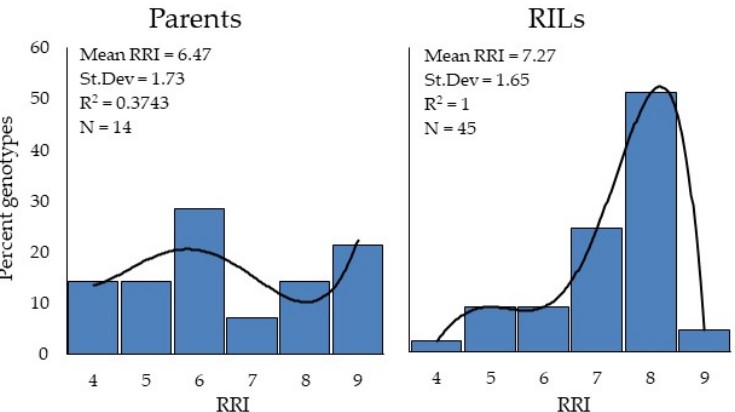

**Figure 10.** Frequency distribution of RRI in parents and F6 selected RIL populations evaluated during 2021–2022. (4 represent highly susceptible and 9 highly resistant).

Similar to the parents (Figure 3), the $F_6$ RILs showed a negative regression trend between RRI and ACI with an $R^2$ value of 0.94, indicating a strong negative relationship (Figure 11). The ACI ranged from 1.4 to 56.7%, whereas RRI ranged from 0.9 to 8.8. The regression equation y = −0.1352x + 7.1650, indicates that with a 13% increase in ACI, the resultant RRI of the RILs declines to 7%.

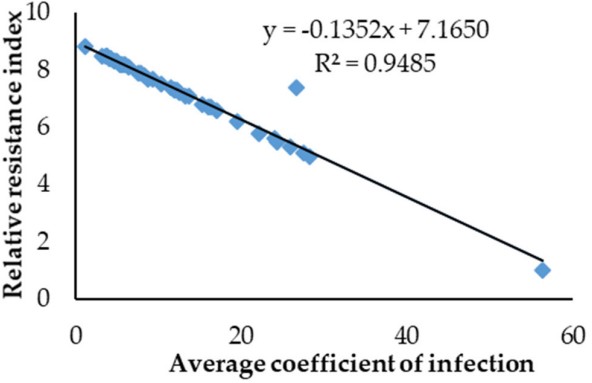

**Figure 11.** Regression analysis between relative resistance index (RRI) and average coefficient of infection (ACI) for yellow rust in RIL $F_6$ population.

Representative RILs from the 45 cross combinations (line × cross) among parents were tested for YR resistance (Table 6). The results showed a diverse disease response in RILs, irrespective of the parental response. So, if two moderate parents are crossed, the resultant RIL could be moderate, resistant, or susceptible. Similarly, the RIL of two resistant parents can also be moderate. An interesting cross combination was AN179 × KS17, in which one parent (AN179) was susceptible and the other parent (KS17) was resistant, resulting in highly resistant RIL with an RRI value of 8.8 ± 0.04. Among the 45 RILs, a higher percentage (50%) showed low ACI (1.4–9.9) with good plant resistance against YR, and 30% RILs showed medium ACI (11–25) with moderate resistance against YR. The remaining 20% RILs showed higher infection with ACI (26–60) with a low level of resistance against YR. Of the 50% RILs showing lower ACI, 40% showed desirable RRI values (>8), whereas 10% of RILs, depicted lower RRI (<5) for YR resistance. These results suggest an increase in resistance of the RILs compared to their respective parents. This may be due to the hybridization of diverse wheat germplasm with unique genetic values.

**Table 6.** Response of 45 RILs $F_6$ wheat populations to yellow rust infections.

| Crosses | Parent Disease Status | Disease Status | Disease Response | ACI | CARPA | RRI |
|---|---|---|---|---|---|---|
| PR123 × PS13 | M × M | M | 40MS | 30.3 | 47.9 | 5.5 ± 0.75 |
| PR123 × PS15 | M × M | R | 10MR | 6.7 | 10.6 | 8.1 ± 0.08 |
| PR123 × PK15 | M × S | S | 40MS | 32.2 | 50.9 | 4.9 ± 1.27 |
| PR123 × K17 | M × R | R | 10MR | 7.1 | 11.2 | 8.2 ± 0.14 |
| PR123 × WD17 | M × R | R | 10MR | 4.4 | 7.0 | 8.3 ± 0.14 |
| PR125 × PS13 | R × M | M | 30M | 15.2 | 24.1 | 7.2 ± 0.48 |
| PR125 × PS15 | R × M | R | 10MR | 3.9 | 6.2 | 8.3 ± 0.17 |
| PR125 × PK15 | R × S | S | 35MSS | 26.5 | 41.9 | 5.3 ± 0.39 |
| PR125 × KS17 | R × R | R | 10MR | 4.6 | 7.3 | 8.3 ± 0.17 |
| PR125 × WD17 | R × R | R | 10MR | 6.0 | 9.5 | 8.2 ± 0.40 |
| PR126 × PS13 | M × M | M | 25M | 16.5 | 26.0 | 6.5 ± 0.77 |
| PR126 × PS15 | M × M | R | 10MR | 4.2 | 6.6 | 8.4 ± 0.10 |
| PR126 × PK15 | M × S | S | 30MS | 21.8 | 34.4 | 5.8 ± 0.24 |
| PR126 × KS17 | M × R | R | 10MR | 4.6 | 7.3 | 8.4 ± 0.02 |
| PR126 × WD17 | M × R | R | 5MR | 3.5 | 5.5 | 8.5 ± 0.02 |
| PR127 × PS13 | R × M | R | 10MR | 6.6 | 10.4 | 8.0 ± 0.19 |
| PR127 × PS15 | R × M | R | 10MR | 4.0 | 6.4 | 8.4 ± 0.14 |
| PR127 × PK15 | R × S | M | 25M | 15.5 | 24.4 | 7.0 ± 0.14 |
| PR127 × KS17 | R × R | R | 10MR | 4.8 | 7.5 | 8.4 ± 0.07 |
| PR127 × WD17 | R × R | R | 10MR | 7.6 | 12.0 | 7.9 ± 0.44 |
| PR128 × PS13 | R × M | S | 20MS | 14.5 | 22.9 | 6.8 ± 0.17 |
| PR128 × PS15 | R × M | M | 5MS | 4.2 | 6.7 | 8.8 ± 0.03 |
| PR128 × PK15 | R × S | R | 10MR | 12.4 | 19.6 | 7.1 ± 0.72 |
| PR128 × KS17 | R × R | M | 20M | 10.5 | 16.6 | 7.6 ± 0.29 |
| PR128 × WD17 | R × R | R | 10RMR | 4.9 | 7.7 | 8.2 ± 0.18 |
| PR129 × PS13 | S × M | M | 30MR | 12.9 | 20.4 | 7.3 ± 0.17 |
| PR129 × PS15 | S × M | M | 30M | 15.9 | 25.2 | 6.6 ± 0.25 |
| PR129 × PK15 | S × S | S | 40MSS | 29.8 | 47.1 | 4.9 ± 0.45 |
| PR129 × KS17 | S × R | R | 10MR | 5.6 | 8.8 | 8.2 ± 0.14 |
| PR129 × WD17 | S × R | R | 10MR | 6.3 | 10.0 | 8.2 ± 0.13 |
| PR130 × PS13 | S × M | S | 30MS | 20.1 | 31.8 | 6.2 ± 0.62 |
| PR130 × PS15 | S × M | M | 25M | 12.8 | 20.2 | 7.3 ± 0.34 |
| PR130 × PK15 | S × S | S | 35MS | 25.9 | 40.9 | 5.1 ± 0.87 |
| PR130 × KS17 | S × R | M | 10M | 7.4 | 11.6 | 8.1 ± 0.17 |
| PR130 × WD17 | S × R | M | 20MR | 9.3 | 14.7 | 7.5 ± 0.20 |
| AN179 × PS13 | S × M | M | 15M | 8.8 | 13.9 | 7.8 ± 0.26 |
| AN179 × PS15 | S × M | M | 20M | 12.3 | 19.4 | 7.7 ± 0.10 |
| AN179 × PK15 | S × S | S | 60S | 56.7 | 89.5 | 0.9 ± 0.40 |
| AN179 × KS17 | S × R | R | 5R | 1.4 | 2.2 | 8.8 ± 0.04 |
| AN179 × WD17 | S × R | R | 20MR | 9.9 | 15.6 | 7.8 ± 0.26 |
| AN837 × PS13 | M × M | S | 20MS | 12.9 | 20.3 | 7.2 ± 0.16 |
| AN837 × PS15 | M × M | S | 30MS | 27.0 | 42.7 | 5.5 ± 0.28 |
| AN837 × PK15 | M × S | M | 25M | 16.7 | 26.3 | 6.7 ± 0.85 |
| AN837 × KS17 | M × R | M | 20M | 12.2 | 19.3 | 7.2 ± 0.21 |
| AN837 × WD17 | M × R | R | 10RMR | 4.7 | 7.5 | 8.3 ± 0.17 |

Key: ACI = average coefficient of index, RRI = relative resistance index, CARPA = country average relative percent attack, YR = yellow rust. Under "disease response" column, from the alpha-numeric codes, the numbers depict the percentage disease severity, e.g., 40MS has 40% disease severity, the alphabetical part is explained as R = resistant, RMR = resistant to moderately resistant, MR = moderately resistant, MS = moderately susceptible, MSS = moderately susceptible to susceptible, and S = susceptible according to Akhtar et al., 2002 [46]. According to vast field experience of the Institute in rust research, two new categories have been added.

## 4. Discussion

Wheat resistance ratings have abruptly changed as a result of the introduction of genetically diverse exotic *Pst.* races into local habitats and their displacement of the clonal *Pst.* races. These *Pst.* races are distinguished by a number of noteworthy characteristics, such as relatively large reductions in resistance in varieties that previously carried effective long-term APR and high production of sexual stage spores (teliospores) [52]. Sexual

recombination has led to the emergence of hypervirulent *Pst.* races. Studies on *Pst.* diversity have reported the emergence of new races and how this has led to evasion of resistance conferred by different Yr genes [14,17]. One of the current study locations, i.e., Kaghan, is vital as it is a natural habitat for YR alternate host *B. vulgarus*, allowing the sexual reproductive cycle required for increasing diversity through recombination and thus the appearance of high virulence *Pst*. Pathotypes.

A diverse gene pool from different geographical locations containing novel resistance genes is a prerequisite to developing YR-resistant wheat cultivars. A large and diverse wheat germplasm pool is being maintained by different working groups such as CIMMYT, the International Center for Agricultural Research in the Dry Areas (ICARDA), Pakistan Agricultural Research Council (PARC), the Center for Agricultural Resources Research (CARR), and Chinese Academy of Sciences (CAS) [53–55]. These germplasms are potential gene pools for tackling different challenges, including the utilization of Yr genes. The currently studied wheat germplasms have been found to be highly diverse as indicated by the wide range of ACI expressions. Studies have shown that the RRI of the plants increases as the ACI decreases [56], which is consistent with the findings from the current study.

This study in the YR hotspot aimed to profile different wheat genotypes using RRI based on ACI under artificial inoculation of a known *Pst.* virulent strain *Pst.* 574232. However, natural infection is also predominant in the region, which may also have infected the experimental plants. The *Pst.* strain used in this study has been found to be avirulent against Yr5, 10, 15, 24, 32, SP, and Tye, whereas virulent against Yr1, 6, 7, 8, 9, 17, 27, 43, 44, and Exp2 [57]. Desirable RRI values have been used as a successful tool for the incorporation of YR resistance in wheat-breeding programs [58]. Three distinct resistance bulks were identified among the tested germplasm, exhibiting complete, moderate, and no resistance against YR. Findings from this study revealed that out of 14, five parental lines were fully resistant against YR, with an RRI value > 8; considered a highly desirable resistance index [46]. In previous studies, nine out of fifty Egyptian wheat RILs have been identified as resistant based on the RRI values above [59]. However, classical breeding approaches are too slow and outpaced by the fast-evolving pathogen [60].

Faster alternates to classical approaches include the development of transgenic wheat lines expressing Yr genes [61]. Therefore, identification of Yr genes/QTL is required for incorporation into speed breeding programs to improve YR resistance among the wheat cultivars and advance lines. Microsatellite markers have been described as markers of choice to identify alleles associated with Yr genes [62]. In the current study, we have found novel alleles at different microsatellite markers unique to different resistance bulks. Moreover, we found some higher frequency alleles within the resistant genotypes bulk. These alleles may be of significant importance for the selection of YR-resistant RILs in successive generations.

Yr18 has been identified as an APR gene, which has conferred stable and durable resistance against YR over the last century [36]. The marker Xgwm295 was reported to be in close proximity to Yr18 [63]. Our results suggest a significant effect of polymorphism at marker Xgwm295 (due to the presence of unique alleles among resistance bulks) on resistance against YR due to its linkage with the Yr18 gene. In previous studies, Yr QTL in bread wheat has been identified at marker Xgwm295 [64]. Furthermore, we have identified four alleles at the Yr18-linked marker "cssfr6" showing a significant correlation with RRI. This is suggestive of the presence of Yr18-based resistance among the studied genotypes. Previous studies have shown a close association of the cssfr6 marker with other resistance genes against leaf rust (Lr34) and powdery mildew (Pm38) as well [65]. These results suggest the effectiveness of cssfr6 in differentiating resistant genotypes from susceptible. The absence of Yr18-linked alleles at cssfr6 and other markers has been shown to result in YR-susceptible Australian wheat genotypes [66]. Two moderate and four resistant phenotype-linked alleles at the csLV34 marker identified in the current study are also known to be in close proximity to the Yr18 gene. Studies in China, Ukraine, and Russia involving Lr34/Yr18/Pm38-linked resistance breeding studies have found a strong linkage

of these genes with the marker csLV34 [67]. The highly polymorphic Yr18-linked markers Xgwm295, cssfr6, and csLV34 may be effective in MAS for resistance breeding in wheat.

The Yr5 and Yr15 associated markers analysis have resulted in the identification of unique alleles at markers Xbarc7 and Xgwm11. In previous studies, alleles Xbarc7 have been associated with the Lr13 gene [68] but found weakly linked with the Yr5 gene on chromosome 2B. No explicit reports of its association with any of the known Yr genes have been found. However, this study has demonstrated that two alleles, i.e., Xbarc7–4 and Xbarc7–16, were frequently found in the moderately resistant bulk, whereas Xbarc7–1 was found among the resistant bulk. Findings from the current study indicate that Xbarc7–1 may be used as a novel marker for the selection of YR-resistant genotypes. The allele Xbrac1014 was significantly associated with RRI in the current study; is moderately linked to the Yr5 genes. In previous studies, marker Xbarc101 has been found in close proximity to the YrSP gene (seedling YR resistance) [69]. Based on findings from this study, the Xbarc1014 allele may be co-located with a potential Yr resistance gene/QTL. Another important Yr gene (Yr15) was found in close proximity to marker Xgwm11 in the current study. Two unique alleles were found at marker Xgwm11 in moderately resistant and resistant genotypes, suggesting its positive association with RRI due to its linkage with the Yr15 gene. The Yr5 and Yr15 have been identified previously as ASR genes providing resistance against all prevailing races of YR in the USA, China, Pakistan, Egypt, and Africa [70–74].

The Yr24/Yr26 has been widely used among Chinese wheat germplasm as a source of resistance against YR [75]. Recently, reports suggest that highly virulent *Pst.* strains have been able to break the Yr26-based resistance in some Chinese advanced lines, with a virulence percentage in the range of 73–98 percent [26,76,77]. The current data from this study have identified the Yr26–linked marker Xwe173 (on chromosome 1B) as having a strong negative association with RRI. Higher infectivity in lines having Xwe173 negative alleles has been reported in RILs of synthetic-derived wheat line Soru#1 [78]. Allele Xwe173–16 identified in the current study may be used as a marker to screen out highly susceptible YR phonotypes linked to the breakage of Yr26-based resistance.

The Yr62 gene was originally identified as a major effect QTL (QYrPI192252.wgp–4BL) on chromosome 4B with the ability to provide HTAP (explaining up to 60% phenotypic variance for YR resistance) [79]. In this study, Yr62 gene-linked allele Xgwm25110 was found to be prevalent among moderate resistant bulk. The slow rusting phenotype in the APR-associated resistant genotypes expressing Yr62 genes is consistent with findings from the current study. Yr62 is another potential resistance gene, especially for areas where hot summers are prevalent.

The data were recorded for the F6 generation of the RILs as by such time about 95% of the genotypes are expected to be homozygous for YR resistance. An upward shift in RRI (8.0) in half of the RILs as compared to 6.0 in one-third of the parental population may be explained by carryover and favorable recombination among potential Yr QTL and genes. Previous studies have found the carryover effect of Yr QTL, e.g., QYr.inra–2BL was found to count for 61% of the resistance phenotype variability among RIL [69]. QYrco.wpg–1B.1, QYrco.wpg–1B.2, and other QTL together resulted in up to 48% disease reduction in the RIL populations [80]. Based on the microsatellite marker analysis, diverse allelic distribution in the parent genotypes and their association with RRI could explain the significant increase in RRI among the resultant RIL population. This suggests that the identified genetic diversity among the parents, including novel alleles associated with variance in resistance phenotypes, is a rich genetic resource and could be used to improve YR resistance in wheat breeding programs.

## 5. Conclusions

The discovery of new alleles confirms the utility of microsatellite markers in differentiating susceptible, moderately resistant, and resistant wheat genotypes. The association and proximity analyses back up reports and field observations on the evolution and establishment of new *Pst.* races, which resulted in Yr26, Yr5, and Yr15 resistance breaks. A number

of associated microsatellite markers having allelic correlations with RRI confirmed the persistent Yr18-based resistance. The enhanced RRI observed in the RIL population could be ascribed to the parents' Yr gene pool and subsequent genetic recombination. Pyramiding of ASR genes such as Yr5, Yr10, Yr15, and Yr26 with APR genes such as Yr18 and Yr62 would offer durable and effective resistance against YR. Data on marker Xbarc101 may indicate the presence of a novel Yr gene or QTL at the same locus. The knowledge base on microsatellite associations generated from this study should form the foundation for long-term YR resistance breeding in wheat.

**Supplementary Materials:** The following supporting information can be downloaded at: https://www.mdpi.com/article/10.3390/agronomy12122951/s1, Table S1: Allele frequencies variations among YR susceptible, moderate, and resistant bulks.

**Author Contributions:** Conceptualization, writing—original draft preparation, M.S.; software, formal analysis, writing—original draft preparation, M.I. and W.A.; data curation, investigation, M.T., S.A., M.N.K. and S.A.J.; methodology, validation, S.J.S., S.Z. and L.S.; visualization, writing—review and editing, F.M., A.Z. and J.L.; supervision, project administration, funding acquisition, H.S. and C.M. All authors have read and agreed to the published version of the manuscript.

**Funding:** Grants from China's National Key Research and Development Program (2017YFD0100804 and 2016YFD0101802), The Agriculture Research System (CARS-03), Anhui Province's University Synergy Innovation Program (GXXT-2019-033), and Jiangsu Collaborative Innovation Center for Modern Crop Production supported this research (JCIC-MCP).

**Institutional Review Board Statement:** Not applicable.

**Informed Consent Statement:** Not applicable.

**Data Availability Statement:** GrainGenes Map Data Report: Wheat, Consensus SSR, 2004: https://wheat.pw.usda.gov/cgi-bin/GG3/report.cgi?class=mapdata&name=Wheat,+Consensus+SSR,+2004 accessed on 15 September 2022; GrainGenes Map Data Report: Wheat, Composite, 2004: https://wheat.pw.usda.gov/cgi-bin/GG3/report.cgi?class=mapdata&name=Wheat,+Composite,+2004 accessed on 18 September 2022; GrainGenes Map Data Report: Wheat, Yr genes and QTL: https://wheat.pw.usda.gov/cgi-bin/GG3/report.cgi?class=mapdata&name=Wheat,+Yr+genes+and+QTL accessed on 25 September 2022.

**Acknowledgments:** The first author expresses their gratitude to the Director and Wheat Breeding Section, Cereal Crops Research Institute (CCRI), Pirsabak, Nowshera, Khyber Pakhtunkhwa, Pakistan, for their assistance in conducting the current research. Authors would also like to extend their gratitude to the National Institute of Genomics and Advanced Biotechnology, Islamabad, Pakistan for allowing access to greenhouse facilities and molecular laboratory equipment.

**Conflicts of Interest:** The authors declare no conflict of interest.

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
