# Peer review of "Molecular Characterization of Diverse Wheat Genetic Resources for Resistance to Yellow Rust Pathogen (Puccinia striiformis)"

_agronomy, doi:10.3390/agronomy12122951_

Round 1

Reviewer 1 Report

Detailed comments and suggestions can be found in the attached PDF file.

However, the manuscript needs editing by a professional English editing service or a native English speaker. I tried to suggest changes or highlight the text showing grammatical errors in many places, but I am not supposed to revise the manuscript for grammatical errors.

Author Response

The authors are highly grateful to the worthy reviewer for his thorough review of their manuscript and for the precious comments and suggestions and comments. The authors believe that revised manuscript is in far better shape now. Attached is the pdf file commented by the reviewer. each of the comment is addressed in the pdf file and the required changes have been incorporated accordingly in the revised manuscript.

Reviewer 2 Report

Kindly find the comments at the attachement.

Author Response

Please find attached point-by-point response to the reviewer's comments. Changes have been made accordingly in the revised manuscript with track changes ON.

Round 2

Reviewer 2 Report

Kindly find the comments at the attachement.

Author Response

Dear reviewer,

We are thankful for your insightful comments. Following is our point-by-point response to your comments and manuscript revised accordingly.

Regards,

Note: Reviewer comments are in plain and blue text and author responses are in bold and italic.

Lines 176-177: The selected 45 F6 RILs (each RIL generated from the original cross combinations) along the respective parents (9 lines and 5 testers) were sown at CCRI Pirsabak in the 1st week of November 2021 using randomized complete block design, with three replications.

Single RIL per population (resulted from each cross combination) were selected,

thus totaling 45.

Please explain what criteria were chosen for 45 F6 RILs. Why these specific ones 45 F6 RILs were chosen? Why only single RIL per population?

Author Response: The criteria for RIL selection and generation advancement have now been explained in the “Materials and Methods” section at lines 175-177. Further explanation on use of F6 RILs has been added under “Discussion” at lines 577-578. Moreover, “single RIL per population” has been misquoted in our first response letter. In fact, the data were recorded on selected single population.

Line 285: Table 3. Response of wheat parents to yellow rust infections.

Author response: This is based on Akhtar et al. 2002. The reference has now been added in the footnotes of table 3 and table 6.

Please describe more detailed, not only adding reference eg.

TR       =          Trace severity of resistant type infection

10MR =          10 percent severity of a moderately resistant type infection

50S      =          50 percent severity of a susceptible type infection

Author Response: The detailed explanations of the givens terms have now been added in the footnotes of Table 6.

Line 380: The percentage values show amount of the ancestral intermixing in each resistance bulk.

Please explain more clearly what does it mean amount of the ancestral intermixing?

Authors response: This has now been rephrased in the legends of Figure 6.
